# Double-Blind, Randomized, Placebo-Controlled Study on hzVSF-v13, a Novel Anti-Vimentin Monoclonal Antibody Drug as Add-on Standard of Care in the Management of Patients with Moderate to Severe COVID-19

**DOI:** 10.3390/jcm11112961

**Published:** 2022-05-24

**Authors:** Prasenohadi Prasenohadi, Erlina Burhan, Sri Dhunny, Wahyuningsih Suharno, Paul Wabnitz, Yoon-Won Kim, Nicola Petrosillo

**Affiliations:** 1Department of Pulmonology and Respiratory Medicine, Faculty of Medicine, Persahabatan Hospital, Universitas Indonesia, Jakarta 13230, Indonesia; praseno@gmail.com (P.P.); erlina_burhan@yahoo.com (E.B.); 2Department of Pulmonology, Pasar Minggu General Hospital, South Jakarta 12550, Indonesia; dhunyatasasri@yahoo.com; 3Department of Pulmonology, Pertamina Pusat Hospital, Jakarta 12120, Indonesia; wahyusuharno@gmail.com; 4clinPHARMA, Precision Medicine & Clinical Trials, Adelaide, SA 5061, Australia; paul.wabnitz@clinpharma.com.au; 5ImmuneMed, Chuncheon 24232, Korea; ywkim@immunemed.co.kr or; 6Department of Microbiology, Faculty of Medicine, Hallym University, Chuncheon 24252, Korea; 7Head, Infection Prevention & Control—Infectious Disease Service, Foundation University Hospital, Campus Bio-Medico UniCampus University, 00128 Rome, Italy

**Keywords:** COVID-19, anti-vimentin antibody, placebo-controlled, phase II study

## Abstract

Humanized Virus Suppressing Factor-variant 13 (hzVSF-v13), a monoclonal IgG4 antibody against vimentin, was investigated in moderate to severe COVID-19 pneumonia through a Phase II study. Patients were randomized to two different IV doses of the test drug or saline with standard of care. Overall, 64 patients were recruited, and 62 entered the efficacy assessment in the full analysis set. Primary endpoint: The clinical failure rate at day 28 was 15.8% for placebo, 9.1% for low-dose hzVSF-v13 and 9.5% for high-dose hzVSF-v13 (not significant). A trend toward better efficacy was shown in several secondary endpoints, with statistical significance between low-dose hzVSF-v13 and placebo in terms of the rate of improved patients on the ordinal scale for clinical improvement (OSCI): 90.0% vs. 52.63% (*p* = 0.0116). In the severe stratum, the results of low-dose hzVSF-v13 vs. placebo were 90.0% and 22.2% for OSCI (*p* = 0.0092), 9 days and 14 days for time to discontinuation of oxygen therapy (*p* = 0.0308), 10 days and 15 days for both time to clinical improvement (TTCI) and time to recovery (TTR) and *p* = 0.0446 for both TTCI and TTR. Change from baseline of NEWS2 score at day 28 was −3.4 vs. + 0.4 (*p* = 0.0441). The results propose hzVSF-v13 as a candidate in the treatment of severe COVID-19.

## 1. Introduction

Damage from COVID-19 results from both the SARS-CoV-2 virus and overactive host immune responses. Therapeutic agents that focus solely on reducing viral load or hyperinflammation fail to provide satisfying outcomes in all cases. Although viral and cellular factors have been extensively profiled to identify potential anti-COVID-19 targets, new drugs with significant efficacy remain to be developed. Moreover, the appearance of virus variants of the original strain potentially becoming resistant to antiviral drugs may lead to decreased efficacy of those agents.

Vimentin is the major component of the type-III intermediate filament that maintains cytoplasm architecture and is critical during influenza virus infections, as it facilitates endosomal trafficking and acidification and mediates viral genome penetration into the cytoplasm, as well as propagation of infection [1]. It was reported that cytoplasmic vimentin is translocated to the cell surface, where it interacts directly with the SARS-CoV spike protein during viral infection, serving as a putative coreceptor involved in cell entry of SARS-CoV [2]. Vimentin has been shown [3] to play important roles during infection by viruses from multiple families with different types of genomes (DNA, single-stranded RNA and double-stranded RNA) and replication cycles. There is also considerable diversity in the viral cycle stages that are impacted by vimentin, with reports including early stages, such as binding/entry, fusion or release of virus genome to the cytosol, or later stages, such as replication or assembly. An integrin binding motif in the toxin appears to be important for binding to intact cells and to recombinant vimentin [3]. Extracellular vimentin is implicated, among other activities, in mediating the invasion of host cells by viruses [4], including SARS-CoV. The role of vimentin as a possible cellular target for the treatment of COVID-19 was first proposed by Li et al. (2020) [5]. The significant role played by vimentin in virus-induced infection is well established as follows: (1) vimentin has been reported as a coreceptor and/or attachment site for SARS-CoV; (2) vimentin is involved in viral replication in cells; (3) vimentin plays a fundamental role in both viral infection and the consequent explosive immune-inflammatory response; and (4) a lower vimentin expression is associated with the inhibition of epithelial to mesenchymal transition and fibrosis. Moreover, the absence of vimentin in mice makes them resistant to lung injury. Li et al. [5] postulated that because vimentin has a twofold role in the disease, both in the viral infection and in the associated life-threatening lung inflammation, the use of vimentin-targeted drugs may offer a synergistic advantage as compared with other treatments not targeting vimentin, and those drugs tested in clinical trials may broaden the therapeutic options against SARS-CoV-2. Furthermore, it has been emphasized [6] that surface vimentin acts as a coreceptor between the SARS-CoV-2 spike protein and the cell-surface angiotensin-converting enzyme 2 receptor, and extracellular vimentin binds directly to the viral domain, confirming an influence of extracellular vimentin in SARS-CoV-2 infections. The possible role of vimentin targeting compounds in treating COVID-19 has been suggested [7] by in vitro and in vivo models of SARS-CoV-2 infection with an investigational small molecule (ALDR491). No clinical studies on the role of vimentin-targeting agents on COVID-19 have been reported to date.

Humanized virus suppressing factor-variant 13 (hzVSF-v13) is a monoclonal IgG4 against vimentin expressed on the surface of virus-infected cells. hzVSF-v13 was reported to have both broad-spectrum antiviral activity and anti-inflammatory effect on virus-induced inflammation [8]. In in vitro studies, hzVSF significantly inhibited HBV infection [9]. Moreover, hzVSF inhibited the cell entry of viral preS1 peptides, possibly by altering intracellular vimentin localization, which is important for HBV cell entry [9]. In a woodchuck hepatitis virus (WHV) model, when hzVSF was provided, together with tenofovir (TAF), which is a conventional therapy for treatment of hepatitis infection, the antiviral effect was enhanced with a more profound suppression of WHV markers in serum and liver than when either drug was used alone. Because the TAF-induced antiviral effect was always transient, this emphasizes the significance of the sustained antiviral response mediated by combination treatment in half of the tested animals [10]. The compound also showed a good safety profile in a Phase I clinical trial in healthy volunteers. In a compassionate use of hzVSF-v13 in patients with critical and severe COVID-19 pneumonia performed in Korea, improvement of pneumonia and complete recovery were observed after administration of the drug in four of seven (57.1%) patients with COVID-19 [11], including a patient who required ECMO and intermittent hemodialysis because of severe pneumonia and IgA nephropathy-induced end-stage renal disease [8].

Our aim was to investigate the efficacy of hzVSF-v13 in moderate to severe COVID-19, as well as its safety profile, through a proof-of-concept study.

## 2. Materials and Methods

### 2.1. Study Design

This was a multicenter, randomized, double-blind, dose-finding, placebo-controlled, parallel group, Phase II clinical study. The study was compliant with GCP (good clinical practice), as well as with the Declaration of Helsinki and subsequent guidance. The study was registered at ClinicalTrials.gov as NCT 04679415.

Those who voluntarily provided a written consent to participate in this clinical study at the screening visit (Visit 1, within 4 days prior to administration of the investigational product) among those who had been diagnosed with SARS-CoV-2 virus infection by RT-PCR test were considered as potential subjects for this study. Only the patients who completed the final eligibility assessment at the baseline visit (Day 1, Visit 2) after the screening test were randomized to the hzVSF-v13 or placebo and standard of care (SoC) combination group (Study Groups A, B, and C) at a ratio of 1:1:1 with severity as a stratification factor. SoC was administered to all groups during the clinical study. An independent data safety monitoring board (DSMB) monitored the safety of subjects and evaluated risk/benefit.

Efficacy and safety assessments were performed at the baseline visit (Day 1, Visit 2), Day 3 (Visit 3), Day 7 (Visit 4), Day 14 (Visit 5), Day 21 (Visit 6), and Day 28 (Visit 7) according to the scheme reported in Figure 1.

### 2.2. Patients

Patients included were adults aged at least 18 years at screening, with diagnosis of moderate or severe COVID-19 by RT-PCR test within 7 days prior to screening, with pneumonia confirmed by radiographic test (CT and/or X-ray) within 5 days prior to the screening visit (e.g., ground glass opacity (GGO), crazy-paving pattern or consolidation). Patients were identified as moderate if their oxygen saturation (SpO_2_) was ≥93%, with respiratory rate (RR) ≥ 20/min or pulse rate (PR) ≥ 90 beats/min secondarily. Severe patients were identified as those with SpO_2_ < 93% or PaO_2_/FiO_2_ < 300, with RR ≥ 30/min or PR ≥ 125 beats/min secondarily. For subjects whose SpO_2_ was not available at screening because they were already using an oxygen mask, severity was classified according to the investigator’s judgement. All patients had to provide a written informed consent prior to any study procedure.

Patients were excluded if they had a clinically significant history of hypersensitivity reactions to the components of the study drugs or to other drugs, including NSAIDs or antibiotics. Also excluded were patients with pneumonia resulting from causes other than SARS-CoV-2 infection, as those with severe heart failure (NYHA Class III or higher), pregnant women, and men and women of childbearing potential not agreeing to use clinically appropriate methods of contraception from day 1 until 120 days after the last administration of the test product. Additional exclusion criteria were schedule to organ transplantation; ALT or AST ≥ 5 times the upper limit of normal (ULN); eGFR < 30 mL/min/1.73 m^2^; platelet count < 50,000/mm^3^; positive for HBV, HCV or HIV; and/or those patients vaccinated against COVID-19 within 30 days prior to the screening visit. Also excluded were those patients who had been determined to be ineligible to participate in the clinical study according to the investigator’s medical opinion or subjects who planned to be transferred to another hospital within the study period.

### 2.3. Randomization and Masking

The patients were randomized in three groups to receive a low dose of the test product (group A), a high dose (group B) or placebo (group C) to be administered by slow intravenous infusion. A statistician not involved in the study prepared a randomization table by a block randomization method stratified by severity (moderate and severe). All investigators and patients remained blinded during the whole study duration, except for the personnel performing the study drug dilution. The randomization number of each subject consisted of 5 letters and digits (R-XX-XX)—the first letter R indicates “randomization,” the following two digits correspond to stratified randomization (severity at screening; 01: moderate; 02: severe), and the last two digits of the randomization number correspond to the randomization order (e.g., R-01-01, R-02-01).

### 2.4. Interventions

The doses and duration of administration for each group were as follows:

Study groupsGroup A: (low-dose hzVSF-v13): Standard of care (SoC) + loading dose of hzVSF-v13 200 mg at Day 1 (D1), maintenance dose of hzVSF-v13 100 mg at D3 and D7.Group B: (high-dose hzVSF-v13): SoC + loading dose of hzVSF-v13 400 mg at D1, maintenance dose of hzVSF-v13 200 mg at D3 and D7

Control groupGroup C: (placebo): SoC + 3 doses of the placebo (normal saline) at D1, D3 and D7.

The doses of the study drug were chosen based on previous investigations, including pharmacodynamics in animal models, animal toxicity, safety and pharmacokinetics from a Phase I study in healthy subjects, as well as a compassionate use for treatment of COVID-19 in Korea. The test product was provided as a clear solution in vials containing the active ingredient, hzVSF-v13, 40 mg/mL × 5 mL (200 mg/vial) stored at 2–8 °C, protected from light. Placebo was provided as normal saline solution. The test product was administered after diluting the vials in 0.9% NaCl 100 mL and filtering through a 0.2-micron inline filter. hzVSF-v13 or placebo was administered intravenously for approximately 30 (±10) min.

All patients were given standard of care (SoC) according to the Indonesian “Management Guidelines for COVID-19” [12], with the “Severe Coronavirus Disease 2019 (COVID-19) Infection Patient Care Recommendations” [13,14] and the local practice.

### 2.5. Outcomes

The primary efficacy endpoint was the clinical failure rate at Day 28, defined as death, respiratory failure (patient intubated) or patient in the intensive care unit (ICU).

Secondary efficacy endpoints included (1) changes from baseline in the ordinal scale for clinical improvement (OSCI 8-point scale [13] at Day 7, Day 14 and Day 28); (2) time to discontinuation of oxygen therapy after investigational product administration; (3) time to recovery (TTR) (time to achieve a score of 0 to 3 on the OSCI 8-point scale) after investigational product administration (days); (4) time to clinical improvement (TTCI) (2-point decrease from the baseline score) (days); (5) changes from baseline in PaO_2_/FiO_2_ at Day 7, Day 14, Day 21 and Day 28; and (6) changes from baseline in the National Early Warning Score 2 (NEWS2) at Day 7, Day 14, Day 21 and Day 28.

Antiviral assessment measured the proportion of patients with viral negative conversion in respiratory samples (oropharyngeal and nasopharyngeal smears) by RT-PCR analysis the day after administration of each investigational product (Day 2, Day 4 and Day 8) and one week after the last administration at Day 14.

Cytokine assessment included changes from baseline in IL-1β, IL-6, TNF-α and MCP-1 values by ELISA analysis the day after administration of each investigational product (Day 2, Day 4 and Day 8) and one week after the last administration at Day 14.

The safety assessments were standard and included recording adverse events, vital signs, 12-lead ECG and laboratory tests (hematology, blood coagulation and chemistry, as well as urinalysis), as well as by physical examinations throughout the study period.

### 2.6. Sample Size

No prior evidence was available on the effect of hzVSF-v13 on COVID-19. We assumed that hzVSF-v13 administration would reduce the incidence of failure in the study groups by up to half (E = 50%) compared to that in the control group. With a type-1 error (alpha) of 0.15% and power of 80%, at least 21 patients were required in each group. In total, at least 63 patients were needed.

### 2.7. Statistical Analysis

The analysis populations were defined by standard methods as full analysis set (FAS), per-protocol set (PPS) and safety set. Efficacy assessment analysis was carried out primarily in the FAS and secondarily in the PPS. Primary endpoint clinical failure rate was descriptively analyzed with 85% confidence intervals (CIs) for each study treatment difference versus placebo + SoC with the corresponding odds ratio (OR). In addition, a logistic regression model was applied to assess the pairwise comparison of each hzVSF-v13 + SoC group versus placebo + SoC, adjusting for the stratification factor (moderate vs. severe), including OR and two-sided 85% CIs from the logistic regression.

Descriptive statistics and a two-sample t-test or the Wilcoxon rank-sum test, depending on whether the normality assumption was satisfied, were applied for the following secondary endpoints: changes from baseline in the WHO OSCI 8-point scale at Day 7, Day 14 and Day 28; changes from baseline in PaO_2_/FiO_2_ at Day 7, Day 14, Day 21 and Day 28; and changes from baseline in the NEWS2 score at Day 7, Day 14, Day 21 and Day 28. A Kaplan–Meier curve with median time and 95% confidence interval by treatment group was generated for time to recovery. Additional efficacy analyses were performed in the severe stratum. Antiviral activity and cytokine changes were analyzed by standard methods.

## 3. Results

### 3.1. Patient Enrollment and Demographics

The study was conducted at three sites located in Indonesia, and a total of 64 patients were randomized: 22 patients in group A were administered SoC + low-dose hzVSF-v13; 23 patients in group B were randomized to SoC + high-dose hzVSF-v13 and 21 patients were included in the FAS; 19 patients in group C were given SoC + placebo (normal saline). Patient disposition is summarized in Figure 2. Recruitment took place between 27 January 2021 (first patient in) and 27 July 2021 (last patient in). The last patient out was on 19 August 2021, and DB lock took place on 5 October 2021.

The demographic characteristics of the patients included are summarized in Table 1. The patients were 28 to 73 years old, and there was no significant difference in mean age among the three groups. All included patients were Indonesian. A slightly higher percentage of patients aged <50 years was recorded in the placebo group (about 47%) compared to the low-dose (41%) and high-dose hzVSF-v13 (43%) groups. There was a prevalence of male patients in all groups, with the high-dose hzVSF-v13 group being the most balanced between genders. Body mass index (BMI) ranged between 18.7 and 37.5 and was also well balanced among the three treatment groups. A total of 62 patients were admitted to treatment: 19 (10 moderate and 9 severe) in the placebo group, 22 (11 moderate and 11 severe) in the low-dose hzVSF-v13 group and 21 (12 moderate and 9 severe) in the high-dose hzVSF-v13 group.

Concurrent diseases were reported in 74.2% of patients; among the risk factors for severe COVID-19, kidney diseases were present in 17.4%, diabetes mellitus in 25.8%, obesity in 8.1% and cardiac disorders in 17.7% of patients. The proportion of those factors was not significantly different among treatment groups. Hypertension was recorded in 10.5% of patients in the placebo group, 40.9% in the low-dose hzVSF-v13 group (Fisher’s exact test, *p* = 0.0385 vs. placebo) and 33.3% in the high-dose hzVSF-v13 group (*p* = 0.1328).

### 3.2. Primary Endpoint

Clinical failure rates in the FAS are shown in Table 2: 15.8% for the placebo group, 9.1% for the low-dose hzVSF-v13 group and 9.5% for the high-dose hzVSF-v13 group. Differences were not significant; the failure rate was lower than expected, and the study was underpowered with respect to the primary endpoint. Table 2 also reports the results of the primary outcome broken down by event, reflecting those of the composite endpoint. The analysis in the PP population confirmed the same trend of better results in the two hzVSF-v13 dose groups compared to the placebo group, although the differences were still not significant.

Clinical failure rates in the severe stratum (FAS) are shown in Table 3: 33.3% for the placebo group, 9.1% for the low-dose hzVSF-v13 group and 11.1% for the high-dose hzVSF-v13 group. Clinical failure in low- and high-dose hzVSF-v13 groups was thus 72.7% and 66.7% lower than in the placebo group, respectively, but was not statistically significant. The trend of better efficacy was maintained in the severe stratum, but the differences were not significant.

### 3.3. Secondary Efficacy Endpoints

Changes from baseline in the WHO OSCI are summarized in Table 4 and in Figure 3. No difference was observed at day 7, but at day 14, any improvement (≥1 point decrease from baseline) was achieved by 52.6% of patients in the placebo group and by 90.0% and 71.4% of patients in the low-dose and high-dose hzVSF-v13 groups, respectively. In the logistic regression with covariate at day 14, the difference between the low-dose and placebo groups was significant (*p* = 0.0116). In the severe stratum, both the baseline OSCI and the time course of changes throughout the study period were very similar to those of the overall FAS. The maximum differences were recorded at day 14 in mean: −0.1 for the placebo group, −1.8 for low-dose hzVSF-v13 group and −1.1 for the high-dose hzVSF-v13 group. Improved patients accounted for 22.2% in the placebo group, 90.0% in the low-dose hzVSF-v13 group (*p* = 0.0092 versus placebo) and 55.6% in the high-does hzVSF-v13 group.

The subjects who discontinued oxygen supplementation were accounted for 13/17 patients (76.5%) in the placebo group, 16/18 (88.9%) in the low-dose hzVSF-v13 group and 15/19 (78.9%) in the high-dose hzVSF-v13 group. The median time to discontinuation of oxygen therapy according to a Kaplan–Meier estimate was 8 days in all treatment groups. The differences were not statistically significant. In the severe stratum, the subjects who discontinued oxygen supplementation accounted for 5/9 patients (55.6%) in the placebo group, 10/11 (90.9%) in the low-dose hzVSF-v13 group and 7/9 (77.8%) in the high-dose hzVSF-v13 group. The median time to discontinuation of oxygen therapy according to a Kaplan–Meier estimate was 14 days in the placebo and in the high-dose treatment group, whereas it was 9 days in the low-dose hzVSF-v13 group (*p* = 0.0308 vs. placebo). Details are found in Table 5.

Subjects who recovered in the FAS accounted for 68.4% of patients in the placebo group, 81.8% in the low-dose hzVSF-v13 group and 81.0% in the high-dose hzVSF-v13 group. Median TTR in the FAS was 9.0, 9.0 and 8.0 days for the placebo, low-dose and high-dose treatment groups, respectively, and the difference was not significant, although there was a trend toward better efficacy in the hzVSF-v13 groups compared to the placebo group. The stratified analysis of severe patients is summarized in Table 6 and the Kaplan–Meier curve in Figure 4: recovery was recorded in 55.6% of patients in the placebo group, 81.8% in low-dose hzVSF-v13 group and 66.7% in the high-dose hzVSF-v13 group. TTR was 10 days in the low-dose group compared to 15 days in the placebo group: hazard ratio (HR) 3.14 (85% CI 1.32, 7.51, *p* = 0.0446). On the contrary, no significant difference was observed between the high-dose hzVSF-v13 group and the placebo group. An additional analysis was performed, excluding the subjects with a WHO OSCI score of 3 at baseline. The subjects who recovered accounted for 64.7% of patients in the placebo group and 77.8% in both the low- and high-dose hzVSF-v13 groups. Median TTR was 13 days for the placebo group and 9 days for both the low- and high-dose hzVSF-v13 groups.

Subjects in the FAS with clinical improvement (2-point decrease from baseline) accounted for 68.4% of patients in the placebo group, 77.3% in the low-dose hzVSF-v13 group and 80.9% in the high-dose hzVSF-v13 group. TTCI was 10, 10 and 11 days for the placebo, low-dose and high-dose treatment groups, respectively. In the stratified analysis for severe patients (Table 7), subjects with clinical improvement accounted for 55.6% of patients in the placebo group, 81.8% in the low-dose hzVSF-v13 group and 66.7% in the high-dose hzVSF-v13 group. Median TTCI was 10 days for the low-dose hzVSF-v13 group and 15 days for the placebo group: HR 3.18 (85% CI 1.31, 7.69, *p* = 0.0446). No significant difference was present between the high-dose hzVSF-v13 group and the placebo group.

Median PaO_2_/FiO_2_ at baseline was 186 for the placebo group, 153 for the low-dose hzVSF-v13 group and 266 for the high-dose hzVSF-v13 group. The parameter had already increased in both the low- and high-dose hzVSF-v13 groups at Day 7, whereas it did not change in the placebo group. The trend of increased change from baseline was maintained in the low-dose group at Day 14, but the difference was not significant. Starting from Day 21, there was a trend toward normal index values in all groups. The analysis of the severe stratum confirmed a similar trend, but the difference was not significant.

In FAS, the NEWS2 score at baseline was 4.4 for the placebo group, 4.1 for the low-dose hzVSF-v13 group and 4.0 for the high-dose hzVSF-v13 group. No difference was observed among the three groups on Day 7. On Day 14, there was a trend toward a larger decrease in both test groups, with a score of 4.7 for the placebo group, 1.9 for the low-dose hzVSF-v13 group and 2.3 for the high-dose hzVSF-v13 group. The improving trend was maintained at day 21, but the difference was not significant. At Day 28, the NEWS2 score was 3.1 for the placebo group, 1.4 for low-dose hzVSF-v13 group (*p* = 0.0570 vs. placebo at the ANCOVA) and 1.7 for the high-dose hzVSF-v13 group. 

Changes from baseline in the NEWS2 score in the severe stratum are summarized in Table 8. There was no change or even an increase in NEWS2 score in the placebo group during the study period. On the contrary, the score decreased gradually at all time points in the hzVSF-v13 groups, with a statistically difference in change from baseline between the low-dose and placebo groups at Day 14 and Day 28. At Day 28, the NEWS2 score in severe stratum was 5.3 for the placebo group, 1.1 for the low-dose hzVSF-v13 group (*p* = 0.0153 vs. placebo at the ANCOVA) and 2.7 for the high-dose hzVSF-v13 group.

The rate of positivity to SARS-Cov-2 gradually decreased in all groups. The assessment of conversion to negative in oropharyngeal or nasopharyngeal specimens did not evidence any better effect of the study drug over the placebo (data not shown). In the cytokine assessment, IL-1β and IL-6 revealed in-treatment and post-treatment values in the high-dose and low-dose hzVSF-v13 groups generally lower than those of placebo values, but the differences were not significant. TNF-α and MCP-1 showed the same trend in all treatment groups (data not shown).

### 3.4. Safety

Safety was good in all treatment groups. Serious adverse events (SAEs) were experienced by four patients in the placebo group, two in the low-dose hzVSF-v13 group and three in the high-dose hzVSF-v13 group. No SAE was related to the study treatment. There were three treatment-related adverse events in the placebo group: one case of nausea, one case of pruritus and one case of pain at the injection site. In the low-dose hzVSF-v13 group, there were three adverse events judged as related to the study treatment: two increases in liver enzymes and one case of anemia. In the high-dose group, three related adverse events were observed, including one liver injury, one case redness and one case of pain at the injection site. No infusion-related reaction was serious. 

No other clinically meaningful change was observed in laboratory tests, apart from the few cases reported as adverse events. There was no abnormal clinically significant ECG finding either at day 7 or at day 28 compared to baseline.

## 4. Discussion

In the present study, clinical failure at day 28 was 15.8% in the placebo group, which is lower than expected. This result may have been due to better knowledge of COVID-19 in 2021, when the study was performed, than in 2020, when it was planned, with availability in 2021 of a proper SoC and appropriate care. Lower aggressivity of coronavirus variant(s) vs. the original virus may have also played a role. As a consequence, our study was underpowered for its primary endpoint and showed only a trend toward better efficacy of both hzVSF-v13 doses, without statistically significant differences among the treatment groups.

Nevertheless, our study confirms a trend of better efficacy of hzVSF-v13 plus SoC compared to placebo plus SoC, with the low-dose hzVSF-v13 treatment being marginally more effective. A trend toward a better effect of the investigational product was evident in the primary endpoint clinical failure: 72.7% and 66.7% lower than placebo group for the low- and high-dose treatment groups, respectively, even if not significant due to the paucity of events. A trend toward better effect was observed also in the secondary endpoints, clinical improvement in OSCI score, recovery rate and NEWS2 score. The percentage of improved patients at Day 14 was significantly higher in the low-dose hzVSF-v13 group compared to that of the placebo group. Subgroup analysis in the severe stratum showed that secondary endpoints TTCI, TTR, OSCI score, discontinuation of oxygen supplementation and NEWS2 score in the low-dose group were significantly better than those of the placebo group, postulating hzVSF-v13 as a candidate for the treatment of severe COVID-19. Beneficial effects of the drug at dose in the range of 50–200 mg were previously observed in a compassionate, uncontrolled use [8]. In planning this study, the lower dose was expected to show efficacy in COVID-19, and the higher dose was investigated to see whether there was further room for improvement. As a result, the high-dose hzVSF-v13 group showed similar or somewhat inferior efficacy compared to the low-dose hzVSF-v13 group. Risk factors for severe COVID-19 did not contribute to the result, as the only significant difference was observed in the ratio of patients with hypertension, which was higher in the low-dose hzVSF-v13 group compared to placebo group. It seems that the low dose reached a plateau of efficacy, and no dose–response relationship above the low-dose hzVSF-v13 group was observed in this study; thus, no further benefits are expected at higher doses. Additionally, it has been generally observed for other biologics that low doses of an mAb can elicit a greater immune/pharmacodynamic response compared to a high dose of the same mAb. The mechanism behind this observation is not clear, and this phenomenon is not an unusual finding [15].

Among the treatments for severe COVID-19 that have been investigated in randomized, controlled clinical trials, dexamethasone was found to reduce mortality among patients receiving mechanical ventilation [16,17]. Remdesivir shortened time to recovery but did not have a significant effect on mortality [18]. Anti-SARS-CoV-2 monoclonal antibodies that target the spike protein have been shown to have clinical benefit in treating SARS-CoV-2 infection, but the appearance of new variants has markedly reduced in vitro susceptibility to most of them; therefore, according to the NIH, such a regimen is not expected to provide clinical benefit for patients [19].

It is presumed that a significant number of patients infected with the delta variant were enrolled in this clinical trial. Nevertheless, hzVSF-v13 showed a significant effect in severe patients. This is thought to be related to the mechanism of the drug binding to vimentin rather than acting directly on the viral substance.

Our study is the first to show the role of a vimentin-targeted drug in the management of COVID-19. Our results seem to confirm that hzVSF-v13 may offer a synergistic advantage as compared with other treatments that do not target vimentin, as vimentin has a twofold role in the disease, both in viral infection and in the associated life-threatening lung inflammation. The results of our study seem to confirm the effect of hzVSF-v13 in the management of COVID-19 independently of its antiviral effect. Its efficacy on clinical parameters vis-à-vis its preventive effect on cytokine storm suggests hzVSF-v13 as a candidate for treatment of patients with severe disease.

The main limitation of our study is the small number of patients included, which prevented us from reaching firm conclusions on the efficacy of the tested drug in COVID-19 pneumonia. Nevertheless, our data constitute proof of concept, providing the necessary premise for further studies to confirm the role of hzVSF-v13 in the management of severe COVID-19. Moreover, our results provide information on the optimal dose to investigate hzVSF-v13 in pivotal trials.

## Figures and Tables

**Figure 1 jcm-11-02961-f001:**
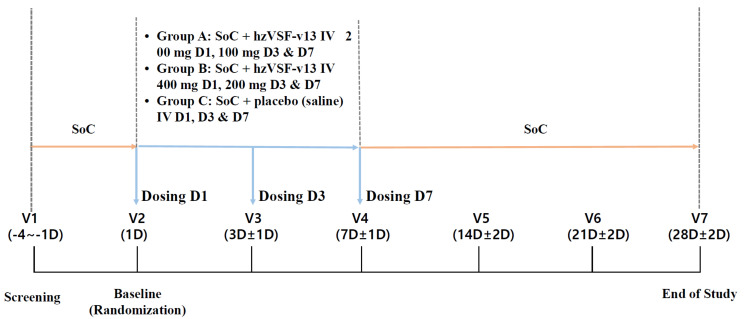
Study design (SoC = standard of care, D = day).

**Figure 2 jcm-11-02961-f002:**
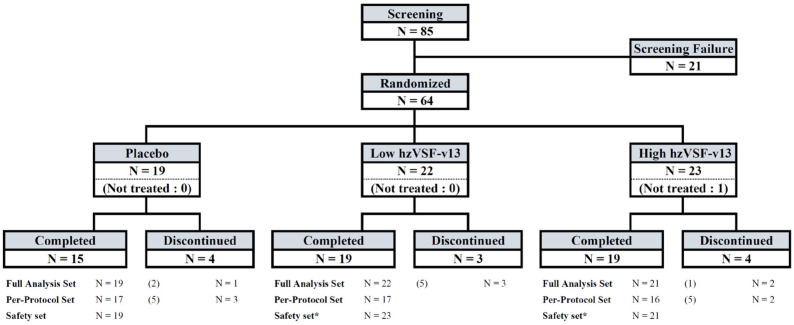
Patient disposition.

**Figure 3 jcm-11-02961-f003:**
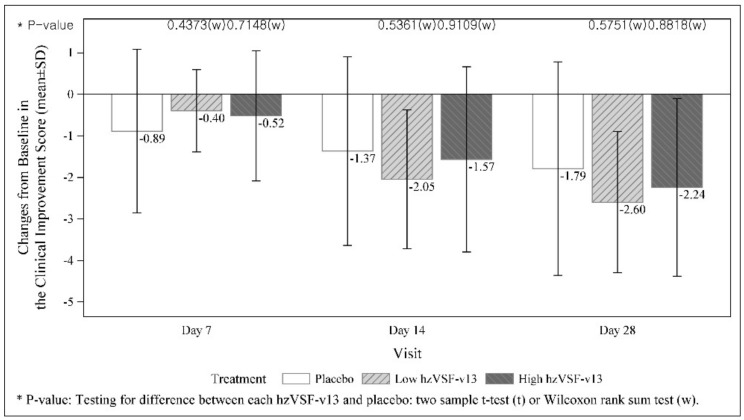
Change from baseline in the ordinal scale for clinical improvement score (OSCI, WHO 2020).

**Figure 4 jcm-11-02961-f004:**
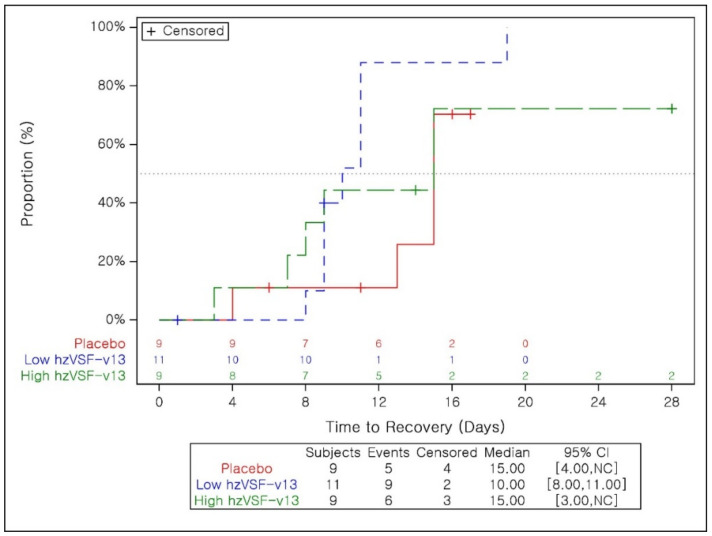
Kaplan–Meier curve for time to recovery (severe stratum).

**Table 1 jcm-11-02961-t001:** Demographic characteristics of the patient population.

Characteristic		hzVSF-v13 Low Dose (N = 22)	hzVSF-v13 High Dose (N = 21)	Placebo (N = 19)
**Age (years)**	Mean [min~max]	50.5 [32~70]	50.5 [34~73]	51.5 [28~71]
**Age categories n (%)**	<50	9 (40.9)	9 (42.9)	9 (47.4)
	50≤~<55	7 (31.8)	7 (33.3)	2 (10.5)
	55≤~<60	2 (9.1)	2 (9.5)	4 (21.1)
	≥60	4 (18.2)	3 (14.3)	4 (21.1)
**Gender n (%)**	Male	14 (63.6)	11 (52.4)	13 (68.4)
	Female	8 (36.4)	10 (47.6)	6 (31.6)
**Body mass index (BMI) (kg/m^2^)**	Mean [min~max]	27.5 [18.7, 36.7]	26.5 [20.2, 37.1]	26.0 [19.8, 37.5]
**BMI categories n (%)**	18.5~24.9	6 (27.3)	9 (42.9)	9 (47.4)
	25.0~29.9	10 (45.5)	8 (38.1)	6 (31.6)
	≥30.0	6 (27.3)	4 (19.0)	4 (21.1)
**N patients**	Admission at enrollment	22 (100)	21 (100)	19 (100)
**COVID-19 severity n (%)**	Moderate	11 (50.0)	12 (57.1)	10 (52.6)
	Severe	11 (50.0)	9 (42.9)	9 (47.4)

**Table 2 jcm-11-02961-t002:** Primary outcome as composite endpoint and by event at Day 28, as well as logistic regression analysis. Full analysis set (FAS).

Group	hzVSF-v13 Low Dose (N = 22)	hzVSF-v13 High Dose (N = 21)	Placebo (N = 19)
**Patients with clinical failure at Day 28 n (%)**	**2 (9.1)**	**2 (9.5)**	**3 (15.8)**
Crude Odds Ratios (ORs)	0.53	0.56	
85% Confidence Interval (CI)	[0.13, 2.16]	[0.14, 2.28]	
*p*-value	0.5180	0.5533	
**Death n (%)**	**2 (9.1)**	**2 (9.5)**	**3 (15.8**)
Crude ORs	0.53	0.56	
85% CI	[0.13, 2.16]	[0.14, 2.28]	
*p*-value	0.5180	0.5533	
**Respiratory failure n (%)**	**0**	**2 (9.5)**	**3 (15.8)**
Crude ORs	0	0.56	
85% CI		[0.14, 2.28]	
*p*-value	0.9515	0.5533	
**Subjects with ICU Admission, n (%)**	**0**	**2 (9.5)**	**3 (15.8)**
Crude ORs	0	0.56	
85% CI		[0.14, 2.28]	
*p*-value	0.9515	0.5533	

**Table 3 jcm-11-02961-t003:** Clinical failure: stratified analysis in severe patients at Day 28. Full analysis set (FAS).

Group	hzVSF-v13 Low Dose (N = 11)	hzVSF-v13 High Dose (N = 9)	Placebo (N = 9)
**Patients with clinical failure** **at Day 28 n (%)**	1 (9.1)	1 (11.1)	3 (33.3)
Crude ORs	0.20	0.25	
85% CI	[0.03, 1.24]	[0.04, 1.57]	
*p*-value	0.2033	0.2768	

**Table 4 jcm-11-02961-t004:** Changes from baseline in the WHO ordinal scale for clinical improvement (OSCI).

	hzVSF-v13 Low Dose	hzVSF-v13 High Dose	Placebo
**Baseline**			
n	22	21	19
Mean (SD)	4.1 (0.7)	4.0 (0.6)	4.1 (0.6)
**Day 14**			
n	20	21	19
Mean (SD)	2.0 (1.9)	2.4 (2.4)	2.7 (2.7)
**Change from Baseline at Day 14**			
n	20	21	19
Mean (SD)	−2.1 (1.7)	−1.6 (2.2)	−1.4 (2.3)
*p*-value (Placebo vs. each hzVSF-v13)	0.5361	0.9109	
*p*-value (Low hzVSF-v13 vs. High hzVSF-v13)		0.5936	
Improved, n (%)	18 (90.00)	15 (71.4)	10 (52.6)
Not Improved, n (%)	2 (10.00)	6 (28.6)	9 (47.4)
**Logistic Regression with Covariate at Day 14**			
Adjusted ORs	11.53	2.48	
85% CI	[2.86, 46.49]	[0.84, 7.32]	
*p*-value	0.0116	0.2266	
**Day 28**			
n	20	21	19
Mean (SD)	1.4 (1.9)	1.8 (2.3)	2.3 (3.0)
**Change from Baseline at Day 28**			
n	20	21	19
Mean (SD)	−2.6 (1.7)	−2.2 (2.1)	−1.8 (2.6)
*p*-value (placebo vs. each hzVSF-v13)	0.5751	0.8818	
*p*-value (low-dose hzVSF-v13 vs. high-dose hzVSF-v13)		0.5036	
Improved, n (%)	18 (90.0)	18 (85.7)	13 (68.4)
Not Improved, n (%)	2 (10.0)	3 (14.3)	6 (31.6)
**Logistic Regression with Covariate at Day 28**			
Adjusted ORs	4.39	2.78	
85% CI	[1.19, 16.21]	[0.86, 9.02]	
*p*-value	0.1032	0.2108	

**Table 5 jcm-11-02961-t005:** Time to discontinuation of oxygen therapy in the severe stratum (NC = not calculated).

	hzVSF-v13 Low Dose	hzVSF-v13 High Dose	Placebo
**Subject with Oxygen Therapy**	11	9	9
**Subject who discontinued the Oxygen Therapy, n (%)**	10(90.9)	7(77.8)	5(55.6)
**Time to Discontinuation of Oxygen Therapy (Days)**			
Median	9	14	14
95% CI	[7.0, 10.0]	[2.0, 27.0]	[3.0, NC]
*p*-value	0.0308	0.6785	
Hazard Ratio	3.33	1.19	
85% CI	[1.42, 7.83]	[0.50, 2.84]	

**Table 6 jcm-11-02961-t006:** Time to recovery in the severe stratum.

	hzVSF-v13 Low Dose	hzVSF-v13 High Dose	Placebo
**Subjects who Recovered, n (%)**	9 (81.8)	6 (66.7)	5 (55.6)
**Time to Recovery (Days)**			
Median	10.0	15.0	15.0
95% CI	[8.0, 11.0]	[3.0, NC]	[4.0, NC]
*p*-value	0.0446	0.5550	
Hazard Ratio	3.14	1.33	
85% CI	[1.32, 7.51]	[0.56, 3.19]	

**Table 7 jcm-11-02961-t007:** Time to clinical improvement in the severe stratum. Clinical improvement is defined as the time from randomization (Day 1) until the first clinical improvement (2-point decreased from the baseline score on the WHO OSCI scale).

	hzVSF-v13 Low Dose	hzVSF-v13 High Dose	Placebo
**Subjects with Clinical improvement n (%)**	9(81.8)	6(66.7)	5(55.6)
**Time to Clinical Improvement (Days)**			
Median	10	15	15
95% CI	[9.0, 16.0]	[8.0, NC]	[4.0, 22.0]
*p*-value	0.0446	0.7514	
Hazard Ratio	3.18	1.25	
85% CI	[1.31, 7.69]	[0.52, 3.01]	

**Table 8 jcm-11-02961-t008:** Changes from baseline in NEWS2 score in the severe stratum ((t) = *t*-test).

	hzVSF-v13 Low Dose	hzVSF-v13 High Dose	Placebo
**Baseline**			
n	11	9	9
Mean (SD)	4.6 (2.5)	5.4 (0.7)	5.1 (1.3)
**Day 14**			
n	10	9	7
Mean (SD)	1.6 (1.5)	2.9 (2.3)	6.0 (4.9)
**Change from Baseline at Day 14**			
n	10	9	7
Mean (SD)	−2.9 (2.8)	−2.6 (2.7)	1.0 (4.3)
*p*-value (Placebo vs. each hzVSF-v13)	0.0377 (t)	0.0612 (t)	
*p*-value (Low hzVSF-v13 vs. High hzVSF-v13)		0.7889 (t)	
**ANCOVA Result at Day 14** (**Placebo vs. each hzVSF-v13)**			
LS Mean Difference (SE)	−4.2 (1.7)	−3.7 (1.9)	
95% CI for Difference	[−7.8, −0.7]	[−7.7, 0.3]	
*p*-value for Difference	0.0235	0.0677	
**Day 21**			
n	10	9	7
Mean (SD)	1.6 (1.6)	2.7 (2.5)	5.4 (5.4)
**Change from Baseline at Day 2**1			
n	10	9	7
Mean (SD)	−2.9 (2.9)	−2.8 (2.8)	0.4 (4.9)
*p*-value (Placebo vs. each hzVSF-v13)	0.0986 (t)	0.1205 (t)	
*p*-value (Low hzVSF-v13 vs. High hzVSF-v13)		0.9273 (t)	
**ANCOVA Result at Day 21** **(Placebo vs. each hzVSF-v13)**			
LS Mean Difference (SE)	−3.7 (1.8)	−3.3 (2.1)	
95% CI for Difference	[−7.6, 0.2]	[−7.7, 1.2]	
*p*-value for Difference	0.0639	0.1416	
**Day 28/End of Treatment**			
n	10	9	8
Mean (SD)	1.1 (1.1)	2.7 (1.9)	5.3 (4.5)
**Change from Baseline** **at Day 28/End of Treatment**			
n	10	9	8
Mean (SD)	−3.4 (3.3)	−2.8 (2.3)	0.4 (4.0)
*p*-value (Placebo vs. each hzVSF-v13)	0.0441 (t)	0.0621 (t)	
*p*-value (Low hzVSF-v13 vs. High hzVSF-v13)		0.6429 (t)	
**ANCOVA Result at Day 28** **(Placebo vs. each hzVSF-v13)**			
LS Mean Difference (SE)	−4.1 (1.5)	−3.1 (1.7)	
95% CI for Difference	[−7.3, −0.9]	[−6.8, 0.5]	
*p*-value for Difference	0.0153	0.0871	

## Data Availability

Not applicable.

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
