# Peer review of "Double-Blind, Randomized, Placebo-Controlled Study on hzVSF-v13, a Novel Anti-Vimentin Monoclonal Antibody Drug as Add-on Standard of Care in the Management of Patients with Moderate to Severe COVID-19"

_jcm, 2022, doi:10.3390/jcm11112961_

Round 1

Reviewer 1 Report

The authors conducted a multicenter, randomized, double-blind, dose finding, placebo controlled, parallel group, Phase II clinical study on hzVSF-v13, a novel anti-vimentin monoclonal antibody drug as add-on standard of care in the management of COVID-19 moderate to severe patients. 

I consider the manuscript worth publishing, given the fact that finding an effective treatment against COVID‐19 is urgently needed. Obviously, additional research is needed on certain subpopulation of patients in which the drug would prove effective, primarily in relation to the use of vimentin-targeted drugs that may offer a synergistic advantage as compared with other treatments not targeting vimentin, therefore broadening the therapeutic options against SARS-CoV-2.

The major strength of the manuscript is that it provides data on the clinical impact of novel treatment with hzVSF-v13 on patients with moderate to severe COVID-19, although statistical significance was not achieved, probably due to small sample size. Although hzVSF-v13 didn’t significantly reduced the clinical failure rate in this cohort, authors showed that hzVSF-v13 could be a candidate in the treatment of severe COVID-19 since the trend to better efficacy was shown in several secondary endpoint with statistical significance between low dose hzVSF-v13 and placebo. Moreover, the appearance of virus variants of the original strain potentially becoming resistant to the antiviral drugs may lead to a decreased efficacy of antiviral agents bringing the treatment focus on such monoclonal antibodies having a twofold role in preventing of disease development.

Title it concise, accurate and relevant.

Abstract adequately summarise the work and conclusions.

Current knowledge and recent developments on the subject are briefly stated.

Methods section clearly narrates what the authors did.

Mayor findings are clear and concise; figures and tables are satisfactory (some changes needed are discussed in the review).

Similar studies in the literature are discussed and limitations presented… an overall well-written manuscript. There is no need for shortening the manuscript.

During the review process I have not noticed elements of plagiarism, fraud or other ethical concerns. Therefore, I consider the manuscript ethically eligible.

The manuscript adheres to the journal's standards and brings valuable and evidence-based information on the possible improvement of care for patients with COVID-19.

However, I would like to point some shortcomings that should be considered before publishing the manuscript.

 Mayor comments:

  1. How do the authors explain the fact that poorer efficacy was shown in the high-dose group?
  2. Can the authors present data on comorbidities in patients given that the severity and outcome of the disease may depend on the presence of one or more risk factors?
  3. BMI was higher in hzVSF-v13 low dose group (Group A) where 45.5% of patients had BMI ≥ 30.0 (compared to 19% in Group B and 21.1 in Group C). That fact should be discussed.
  4. Could the authors comment why any differences were shown between study groups in the cytokine assessment?
  5. Since patients were randomized in groups A, B, and C, the columns in Table 1 should be shown in the same order, therefore moving placebo to the last column. It would be appropriate to present the other results in the same order.

Minor comments:

  1. One of the names of the first author is missing – “Prasenohadi Prasenohadi”(Line 6)
  2. Table titles should start with a capital letter (Results, Lines 241 and 251)
  3. I would suggest use of “in the management of patients with moderate to severe COVID-19” instead of “in the management of COVID-19 moderate to severe patients” (Title, Line 4-5, and onwards)
  4. I would suggest use of “humanized Virus Suppressing Factor‐variant 13 (hzVSF-v13), a monoclonal IgG4 antibody against vimentin,…” instead of “hzVSF-v13, a humanized monoclonal IgG4 against vimentin” (Abstract, Line 20, and onwards)
  5. “in patients with critical and severe COVID‐19 pneumonia performed in Korea, improvement of pneumonia and complete recovery were observed after administration of the drug in 4 out of 7 (57.1%) patients with COVID-19 [11],  including a patient who required ECMO and intermittent hemodialysis because of severe pneumonia and IgA nephropathy-induced end stage renal disease [8]” instead of “on critical and severe COVID-19 pneumonic patients…” (Introduction, Lines 91-95)
  6. Last words of the second sentence of the second paragraph in Results “around 50 to 51 years old” are unnecessary and should be excluded since values of the mean age are shown in Table 1 (see Line 233)

Reviewer 2 Report

the authors are to be commended for conducting a trial in this area - the study highlights the challenges of conducting research where the infective organism has changed in its virulence and medical care has improved in parallel with increased expertise and knowledge. the study result was negative but i wonder if it had been conducted a year earlier would the outcome have been different. Nonetheless as the authors point out in the discussion there remains a need for covid-19 directed therapies particularly in immunocompromised - it would be interesting to conduct this study in such a grouping one where we are struggling at present despite vaccinations.

some edits

line 37should read damage from covid 19

line 80 re write to as having broad spectrum and ..... as an antiinflammatory

line 93 replace especially with notably

line 101 - the study was.... - please reword this sentence 
